# A Research and Innovation Agenda for Zero-Emission European Cities

**Francesco Fuso Nerini [1,2,*]**, **Adriaan Slob [3]**, **Rebecka Ericsdotter Engström [1]** and
**Evelina Trutnevyte [4]**

1 Unit of Energy Systems Analysis, KTH—Royal Institute of Technology, 114 28 Stockholm, Sweden;
  rebecka.engstrom@energy.kth.se
2 Payne Institute, Colorado School of Mines, Golden, CO 80401, USA
3 TNO Strategic Analysis and Policy, NL-2509 Den Haag, The Netherlands; adriaan.slob@tno.nl
4 Section of Earth and Environmental Sciences, Renewable Energy Systems group, Institute of
  Environmental Sciences, University of Geneva, CH-1211 Geneva, Switzerland; Evelina.Trutnevyte@unige.ch
* Correspondence: francesco.fusonerini@energy.kth.se

**Abstract:** The Paris Agreement and SDG13 on Climate Action require a global drop in Green House Gases (GHG) emissions to stay within a "well below 2 degrees" climate change trajectory. Cities will play a key role in achieving this, being responsible for 60 to 80% of the global GHG emissions depending on the estimate. This paper describes how Research and Innovation (R&I) can play a key role in decarbonizing European cities, and the role that research and education institutions can play in that regard. The paper highlights critical R&I actions in cities based on three pillars: (1) innovative technology and integration, (2) governance innovation, and (3) social innovation. Further, the research needed to harmonize climate mitigation and adaptation in cities are investigated.

**Keywords:** cities´ decarbonization; European Union; zero carbon cities; smart cities; circular economy; governance; social innovation

---

## 1. Introduction

Cities account from 60 to 80% of global $CO_2$ emissions, depending on the estimates [1]. In the European Union, around three quarters of the population lives in cities, but this share continues to increase [1]. It is clear that the achievement of a "well below 2 degrees" climate change trajectory, as required by the Paris Agreement [2], will need cities as key actors.

Furthermore, cities are the "melting pots" where decarbonization strategies for energy, transport, buildings, and even industry and agriculture coexist and interact [3]—hence the potential for sectorial integration is especially high. Local expertise, density of infrastructure, and the possibility to leverage economies of scale are some of the many reasons to focus on cities in the European decarbonization challenge. In many cases, cities have also launched decarbonization plans that are more ambitious than the national plans. An example is the Covenant of Mayors programme, originally an EU initiative, which to date organizes 7755 cities in ambitious decarbonization commitments from cities around the world [4].

In this context—where city action is both needed and increasingly taken—it is crucial to examine and understand where and how research and innovation (R&I) is needed to support cities in accelerating decarbonization efforts, and then to plan an R&I agenda accordingly. From the academic community, research and development agendas have been proposed for several aspects of urban sustainability. These include suggestions on the future of urban ecology research [5], an agenda for cities as smart interconnected systems [6], as well as proposed efforts to enhance urban climate change

adaptation [7] and resilience [8]. Further, methodology-specific research agendas to accelerate the transformation to low carbon cities have been proposed for "urban ecosystem modeling" [9], "urban living labs" [10], and future energy analysis of the built environment [11]. Most of these include some aspects on how urban societies can transition into sustainable cities with low or zero climate emissions. However, an overarching R&I agenda centered on how research and innovation institutions can support the decarbonization challenge in cities is missing. The New Urban Agenda [12] spans all spheres for urban sustainable development and stresses the importance of climate change mitigation (and adaptation) actions from several perspectives. However, its focus lies primarily on politics and governance. It emphasizes the importance of science and academia (in particular related to the need for social, technological, digital, and nature-based innovations) but does not guide city actors or the scientific community to specific R&I needs.

Many scientific studies on urban-centered climate change mitigation also analyses or review political, technical, or economic measures to curb urban emissions. One study [13] emphasizes that equitable access to low carbon solutions for all (including low-income) urban populations is important for cities to substantially reduce Green House Gases (GHG) emissions. Another study [14] provides analysis of per capita GHG emission of city dwellers in different parts of the world and stresses the importance of emission inventories as a starting point for effective urban climate mitigation. Another study [15] shows how the diversity of (22 studied) cities lead to diverse collections of solutions to effectively reduce emissions. They argue that acknowledging such specific city characteristics can provide additional policy options for nation states in their climate mitigation efforts. This is especially the case for measures that can be directed towards urban infrastructure development and that go beyond carbon pricing or other broader market instruments.

A decade-old study [16] called for future research to adopt an integrated system perspective, integrating all sources, sinks, and opportunities for infrastructure and technology for carbon management of cities. Such research should account for the potential multiple benefits of climate change mitigation in cities (such as combined mitigation and adaptation actions) and identify efficient urban carbon governance (by ascertaining who can influence the urban carbon mitigation, and by what extent). In recent years, initial assessments of such multiple benefits have been assessed related to selected urban sustainability measures [17].

Another complication is that cities in the European Union are very diverse (in terms of technical context, affordability of low carbon investment, governance, etc.). What works in one city does not necessarily work in another. Approaches to the decarbonization of cities are also diverse. To exemplify such differences, this paper compares three EU cities, one in the North of the European Union (Stockholm), one in the South (Barcelona), and one in the East (Warsaw).

However, to date there is a lack of a holistic view on how R&I actions could evolve in the future to support low- and zero-carbon efforts in very diverse European cities. To overcome this research gap, the aim of this paper is to provide an overview of key areas that will need research and innovation to support the decarbonization challenge in the European Union, and to select a number of actions that are perceived to be critical to achieve zero-carbon cities in the European Union. First, the methods for such assessment are presented. Then, selected R&I actions in cities are categorized into three areas: (1) innovative technology and integration, (2) governance innovation, and (3) social innovation. Finally, the paper explores the holistic challenge of climate action in cities, and the role of diverse actors in such a challenge, with a focus on higher research institutions.

*Diverse Challenges and Low-carbon Solutions in Diverse Cities*

Table 1 presents key decarbonization parameters for three European cities: Stockholm, Barcelona, and Warsaw. It shows how action on decarbonization is motivated and organized differently in the three cities. Two main differences appear: (1) how the city governance powers can influence decarbonization planning in cities. The cities' regulatory powers vary significantly across cities; (2) the approach taken on climate action. This point varies in terms of targeted sectors and focus.

**Table 1.** A comparison of three diverse European cities. Content adapted from a previous study [3].

| Parameter | Stockholm | Barcelona | Warsaw |
|---|---|---|---|
| Population (within city boundaries) | 950,000 (2017) | 1,628,936 (2018) | 1,758,143 (2017) |
| Jurisdiction | Strong mayoral powers regarding buildings, city roads, land use, and water. The city owns most of the land, and gets its financing from income taxes. | The city has strong powers and ownership over public buildings and urban land use. However, it has limited power over the city´s energy supply, and partial powers over the transport infrastructure. | Strong local government policy powers and ownership over public buildings, transport infrastructure, roads, and water systems. |
| Key plans acting on decarbonization | The actions for reducing emissions in Stockholm have been centered on heating, transport, waste, electricity, and gas. The city also has a focus on testing new low-carbon solutions in selected neighborhoods, and to then expand effective ones to the whole city. | Most of the policies that will decrease emissions in Barcelona are not specifically addressed at climate change mitigation, which features as a cross-cutting issue across policies, but rather at improving the local air quality and the livability of the city. | Focus on efficiency, transportation, and public awareness. Behavioral changes were promoted through targeted incentives, which were well received by the local population. |

## 2. Materials and Methods

This paper aims to review the current need for research to support the decarbonization of European cities. The methods used can be summarized as an expert-driven, semi-structured literature search guided by experts in the field, developed in the context of the High-Level Panel of the European Decarbonization Pathways Initiative [3,18].

The steps to arrive at the results presented below are the following:

(1) The experts, composed by the authors of this paper, the members of the High-Level Panel of the European Decarbonization Pathways Initiative [3,18], and other members of the H2020 DialoguE on European Decarbonisation Strategies (DEEDS) project, decided on the categorization of the R&I actions in cities trough facilitated discussion over several meetings. Three pillars were selected to categorize future R&I actions in cities: (1) innovative technology and integration; (2) governance innovation; and (3) social innovation. For each of these pillars key R&I actions for cities to become zero carbon by 2050 are proposed. While there are clear connections in topics in these three pillars (e.g., governance needs social innovation and citizen participation), these pillars were deemed useful to categorize and divide R&I actions.

(2) The authors of this paper did a structured literature search for each of these pillars targeted at (a) capturing the current state of the art in R&I for European cities' decarbonization and (b) identifying key R&I gaps for the decarbonization challenge in the European Union´s cities. The authors of this study did not do a comprehensive literature review of all aspects of decarbonization in cities, but a targeted literature review aiming at capturing points (a) and (b) above. For instance, studies looking at which are the most promising technologies for decarbonizing a sector in cities (e.g., heat) were included, but studies going into detail for a single promising technology (e.g., geothermal heat pumps) were not comprehensively reviewed to limit the scope of the study.

(3) The state-of-the-art and R&I gaps discussed above were presented and refined during several meetings, including all the "experts" defined in point 1, during regular meetings for approximately one year. At each meeting, research priorities were discussed and iteratively refined through additional targeted literature reviews and facilitated discussions.

(4) Consensus was reached on the R&I priorities discussed below.

The resulting R&I actions presented below have neither the scope nor the ambition to provide an exhaustive assessment of all the sectoral and cross-sectoral decarbonization challenges in cities.

They represent a deliberate selection of topics that the experts involved in this paper consider of primary relevance for the design of a successful R&I strategy for decarbonizing EU cities. The R&I actions look at how GHG emissions can be decreased in cities with various actions ("decarbonisation achievements"), with the aim of achieving zero carbon cities by 2050; "zero carbon cities" mean cities that are carbon neutral, encompassing all direct and indirect emissions within their boundaries.

## 3. Research and Innovation Actions for Decarbonizing EU Cities

### 3.1. Key R&I Elements within Innovative Technology and Integration in Cities

As seen in Table 1, cities are heterogeneous across regions. Even within cities, building stocks in different areas differ in energy efficiency and level of digitalization. However, this Section identifies some common aspects on the role that innovative technology and integration can have in the decarbonization challenge. Those are categorized under the broad umbrella of smart cities, circular economy, and innovative technology development. While these three concepts are closely interconnected, they are divided here as they represent three interconnected but independent streams of research in literature.

#### 3.1.1. Smart Cities

The first recurring key concept for the integration of low-carbon technologies in cities is the concept of "Smart Cities". A smart city is a city that is technologically interconnected through a network of sensors, IT platforms, open data, and programs that serve to make life within the city more efficient [3]. Smart city projects are diverse, and range from apps for reporting road defects, to the integration of electric vehicles into the city grid balancing [19]. While smart city concepts are being developed every day, there is a need to continue integrating innovative technologies and Innovation and Communication Technologies (ICTs) in the urban system, and to test those solutions in diverse cities. This could include both different designs and technology options, including from smart thermal grids, multi commodity grids, and mobility-as-a-service measures, but also smart lamp posts or smart bins that reduce consumption of energy. It is especially challenging to develop smart cities with new and innovative infrastructure within existing urban systems. There is a need to connect with the existing, sometimes decades- or even centuries-old infrastructure and building stock. Strategies are needed to overcome the trade-off between replacing existing infrastructure with completely new, and potentially expensive, infrastructure; or integrating less-revolutionary solutions that do not significantly challenge the existing interests and system.

#### 3.1.2. Circular Economy

Circular economy (CE) is another key concept often mentioned to decarbonized cities. CE is related to the concept of smart cities, as the former can help enable the circular economy. While the term is used often, there is yet no consensus on the definition of circular economy [20]. By comparing 114 definitions, one study [21] defines circular economy as: "an economic system that is based on business models which replace the "end-of-life" concept with reducing, alternatively reusing, recycling, and recovering materials in production/distribution and consumption processes, thus operating at the micro level (products, companies, consumers), meso level (eco-industrial parks), and macro level (city, region, nation, and beyond), with the aim to accomplish sustainable development, which implies creating environmental quality, economic prosperity, and social equity, to the benefit of current and future generations." Depending on the implementation of CE, it is estimated that it could have different impacts on EU energy usage and emissions. One study [22] estimates that CE could reduce the global primary energy demand by 5% to 9%. Another study [23] estimates that CE could reduce global and EU primary material consumption by 32% and 52%, respectively, by 2050. CE tends to use different technologies, both mature and innovative. Given the wide array of possible solutions, further research will be needed on the technologies that enable CE and how they interact. Waste management,

digitalization, district heating, and transportation optimization are some of the topics that best relate circular economy with technology [3]. Further, knowledge-sharing on how CE is developed in different cities and countries will be essential to understand differences and capture best practices. This should include not only technical aspects, but also financing, governance, and social engagement practices.

### 3.1.3. Heat, Electricity, and Energy Efficiency Technologies

Finally, more research, innovation, and testing will be needed to understand which (and how) technologies can be used in cities to decrease emissions. The innovation in technologies could be in the technology itself, but also an innovative way of using a mature technology. Furthermore, all of these technologies can be used in conjunction as pieces of smart city and circular economy concepts.

Here, there will be a need to share best practices in building efficiency. In fact, while across the European Union, building efficiency has been rising in time, and the European Union has set the target of having all new buildings nearly zero energy by 2020 [24], most of Europe's existing building stock has yet to be affected by energy performance requirements [25]. Continuous research and innovation will be needed to promote both the refurbishment of existing non-efficient buildings and the design of innovative strategies for near zero-energy buildings [3]. That will also include the design of new smart urban spatial strategies when new cities and quartiers are expanded.

Furthermore, with cities being hotspots of energy demand, R&I is needed to understand cities' roles in the local production of electricity and heat. For local electricity production, solar, bioenergy, waste, and wind sources can be harnessed. As for heat, several renewable heat sources can be integrated. Biomass-based CHP, solar thermal units, and waste-to-energy technologies are some of the most mature technologies currently used. In addition, geothermal energy is currently being investigated for its integration in urban areas [26].

### 3.1.4. Suggested Medium-Term R&I Actions for Innovative Technology and Integration

In Figure 1, some key actions are selected for the R&I on innovative technology and integration for decarbonization of cities in the European Union. The first need is to map and disseminate best practices in technologies and strategies for decarbonization in cities. Many cities are developing new innovative approaches, for example to circular economy, and transmitting the lessons learned is key to upscaling such innovative decarbonization solutions. There is a need to understand how renewable energy, electric mobility, and efficient and smart buildings can be integrated in a single city "organism". Smart city concepts and digitalization can provide the tools for integrating such systems in cities. R&I should also explain how this integration could differ in cities that vary by location, size, existing building stock, and transportation infrastructure. Finally, the European Union should engage in a race to the top in cities by developing a series of zero-carbon living labs, where new zero-carbon urban solutions can be tested and replicated.

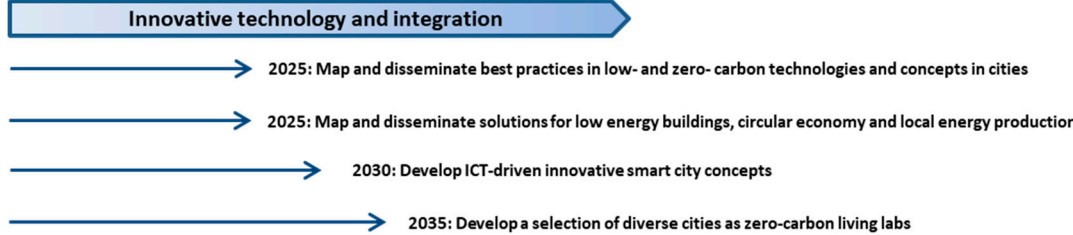

**Figure 1.** Key identified actions for innovative technology and integration for decarbonization in EU cities [3].

### 3.2. Key R&I Elements within Governance Innovation in Cities

The process of shaping the low-carbon transition in cities needs a paradigm shift from formal authority or governmental planning to governance [27,28]. Governance is here defined as "the totality

of actors, rules, conventions, processes, and mechanisms concerned with how relevant risk information is collected, analyzed, and communicated, and how management decisions are taken" [29]. Using the concept of governance thus enables a holistic view to low-carbon transition in cities that cuts across micro-, meso-, and macro-scales, and across all kinds of institutions, responsibilities, rules, and norms in the broadest sense.

A holistic view of governance of low-carbon transition in cities also implies that action is required across multiple sectors and across technical and societal domains. As cities are "melting pots" for various types of infrastructures, integrated urban planning and cross-sector governance are key in order to go beyond sectoral silos and identify the opportunities and benefit from the synergy of coupling various sectors. Fundamental and systemic transition to low- and zero-carbon cities also requires multiple sectors innovating together and involving social innovation (see the next Section). For example, as mentioned above, a circular economy requires reducing, reusing, and recycling waste or recovering materials, whereas some of the waste streams can be used to produce heat, electricity, gas, or fertilizers. A radical transition from personal vehicles with internal combustion engines to new transportation concepts with public transport, car sharing, bicycles, walking, and electric vehicles also cut across the areas of transportation, energy, land-use planning, privacy, safety, and so on. The end-use technologies and social innovations tend to be marginalized so far as compared to technical solutions, especially in the energy supply or transportation domains [30,31]. Citizens are the main users of city infrastructure and the main drivers of consumption of energy, goods, and services. Transition to carbon-neutral cities, thus, should also include citizen-centric innovations, such as diet changes [32], sharing economy [33], device convergence [34], more sustainable forms of consumption [35], and many others (citizen social innovation is discussed in detail in Section 3.3).

Local government still has a pivotal role to play in this multi-actor governance process [36]. They can create a shared, ambitious long-term vision of the low-carbon transition as a way to align the actions of multiple actors towards the same goal [37,38]. When in line with global climate targets but still adapted to the specific local context, visionary concepts are powerful tools because they are endorsed by multiple actors, and hence help mobilize these actors and resources [39]. Examples of such visions are the concepts of a smart city, a circular economy, and a zero-carbon or 100% renewable energy city. Local government can also take a variety of other actions, such as implementing regulatory standards, providing financial incentives, joining public–private partnerships, organizing information and networking events, and so on. In fact, local governments are arguably the right actor to also reach out to the citizens at large, due to their stronger connection to the citizens than national governments, industrial actors, or NGOs [40]. Local governments could, for example, organize processes to find their citizens' low-carbon vision that is broadly legitimized and realistically implementable through public participation processes [39,41]. Furthermore, the analysis of climate activities in global cities showed that new governance schemes are emerging and often involve closer cooperation between public government and private bodies [30]. As illustrated in Table 1, European cities are very diverse. Not only are the challenges or low-carbon solutions different across Europe, but also the regulatory power of cities and the available means to finance or enable financing of low-carbon action. In all cases, the regulatory power of cities is limited and city governments have to interplay with regional, national, and European-level authorities.

Suggested Medium-Term R&I Actions for Governance Innovation

Given this state-of-play in European cities in the context of low-carbon transition, three R&I areas for governance innovation are needed (Figure 2). First, as European cities are diverse in their challenges, solutions, and governance situations, it is important to map the current approaches and best practices to low- and zero-carbon urban governance mechanisms that are used by local governments and other actors. For example, innovations on urban planning strategies are needed for revitalizing or extending existing city neighborhoods and to integrate low-carbon solutions across various sectors from the start, such as renewable energy, low consumption, green areas, and other carbon sinks.

Successful examples of developing broadly appealing low-carbon visions with quantified targets and then new innovations for systemic monitoring of implementation should provide information on what works when and where. It is key that any successful examples and monitoring outcomes are documented in a holistic way that allows for transferability and comparison across many cities in Europe. In this way, lessons learnt and best practices can be transferred from one city to multiple others with similar situations, despite the European diversity. Strategic partnerships between universities, local governments, as well as other stakeholders with the relevant data and tools, could be created to ensure a thorough, data-driven documentation of available governance strategies and their assessment.

Low-carbon transformation in European cities also needs R&I on new tools for financing, incentivizing, initiating new business models, and maximizing information to scale the successful solutions. Many of the current tools are not optimal for holistic, low-carbon measures, because they do not yet cut across sectoral silos, supply and consumption, or technology and society. In terms of multi-actor governance, the current tools also often target one type of actor with their specific powers and responsibilities. R&I could be used to create and assess new types of procurement procedures, public-private partnerships, or public entrepreneurship activities. Citizens can also be further involved as agents of change through measures like participatory budgeting or citizen-run community projects. It is key to understand how the various types of measures interact, from regulation to incentives or information. Universities could contribute here with a collection of independent evaluative evidence for the assessment of these measures.

A far-reaching and fast low-carbon transition in European cities also requires optimization of the role of local governments that are in a network with a multitude of other actors. R&I is, therefore, needed to craft processes of the vertical, multi-level governance that allows the governments to bridge the European Union directives and national policies with local interests, ranging from citizen engagement to the stakes of local companies. The coordination and integration of policy actions and instruments across local, national, and European scales in order to steer their interplay towards low-emission outcomes is key. As citizens of European cities are instrumental to city decarbonization, vertical, multi-level governance processes shall necessarily account for the European citizens' vision of a low-carbon future, low-carbon lifestyles, and social innovations. Through such long-term vision exercises, governments could pilot new ways to leverage resources across various types of public and private actors for productive zero-carbon innovation.

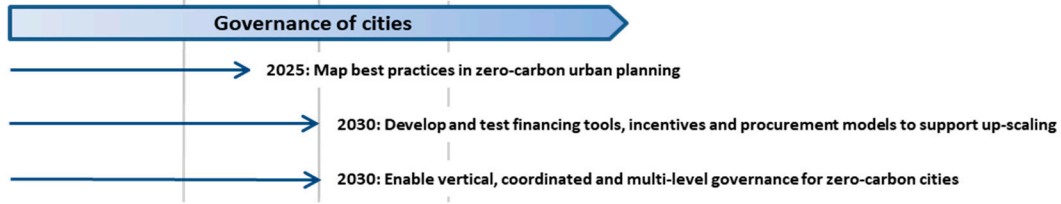

**Figure 2.** Key identified actions on governance innovation for decarbonization in EU cities [3].

### *3.3. Key R&I Elements within Social Innovation*

Social innovation can boost bottom-up decarbonization in cities through local initiatives from citizens or citizen collectives. Social innovations are the result of the action of creative individuals or groups who are able to find innovative solutions to social problems in the community that are not adequately met by the local system. Those, in turn, can also result in governance change and innovation (discussed above). Local initiators act on the social needs and are skilled in finding novel ways (business models, ways of collaboration, funding mechanisms, etc.) to solve the issue. The social process that is initiated in this way, at the same time fosters the local capacity to solve the issue. In this way, social innovation creates new ideas for zero-carbon products, services, new social relationships, or innovative ways of organizing and collaborating that fit in the specific local context. It includes the empowerment of bottom-up initiatives, the embedding of (new) technologies in the

socio-cultural sphere, achieving behavioral and social change, and improving social systems on a local or urban level. With this diversity of topics, social innovation is a rather broad field of research and innovation, and "has become characterized by conceptual ambiguity and a diversity of definitions and research settings" [42]. There seems to be an implicit agreement that an overarching definition of social innovation should contain two "core conceptual elements": (1) a change in social relationships, systems, or structures, and (2) that such change serves a shared human need or goal, or solve a socially relevant problem [42].

Two types of social innovation are of particular interest for the decarbonization challenge of cities: grassroots innovation and social entrepreneurship.

Grassroots innovation is "a network of activists and organizations generating novel bottom-up solutions for sustainable development and sustainable consumption that respond to the local situation and the interests and values of the communities involved" [43]. Grassroots innovations differ from mainstream innovation, as they possess different types of sustainable development and forms, such as cooperatives, informal community groups, social enterprises, and voluntary associations [44].

Social entrepreneurship contains several sub-concepts, which are identified as (a) social value creation, (b) the social entrepreneur, (c) the social entrepreneurship organization, (d) market orientation, and (e) social innovation [45]. The individual, the social entrepreneur, plays a key role in developing innovation that creates (local) social wealth.

Apart from solving local pressing issues, social innovation can also create local jobs. The emphasis on market orientation can differ among social innovation, but in general it is part of social entrepreneurship and grassroots innovations [43,46]. As the social innovations develop further, and the organization becomes more mature, professionalized, or commercialized, it can develop into a business-like organization. Many social innovations shift from a marginal to a commercial organization over time [43,47]. The distinction between "social innovation" and "business innovation" then becomes blurred. Businesses themselves can also develop social innovation [46], which is sometimes seen as a further development of Corporate Social Responsibility [48].

Two key challenges of social innovations in cities are (apart from the many challenges that social innovations are confronted with) the neglect of social innovation in terms of policy making, and the replication and upscaling of social innovations.

Suggested Medium-Term R&I Actions for Social Innovation in Cities

Figure 3 provides some Key identified actions on social innovation for decarbonization in EU cities.

The first one relates to the testing of social innovation strategies in diverse contexts. Many social innovations start on as small scale and are very locally situated, which causes them to generally have a problem in getting attention and recognition from policy makers [44]. On the other hand, social innovation can easily create tension with policy silos and related policies, as they do not keep themselves within the boundaries of defined policy domains while developing solutions for societal problems. Many social innovations operate with this tension between traditional "top-down" policies and "bottom-up" initiatives. In this respect, awareness campaigns for policy makers are needed regarding what social innovation can contribute to decarbonization policies and how social innovation can help to reach decarbonization goals. Research can help to highlight successful examples of the interplay between decarbonization and social innovation and can assist in developing suitable governance models for this interplay.

The second one relates to the scaling up of social innovations. Social innovations are developed in a specific local context for a specific local societal problem. Upscaling within the city or replication in other cities is, therefore, a challenge, and probably not possible for many social innovations in their complete form. Development of business models, cooperation with businesses and public authorities, and targeted replication and upscaling strategies for the (core elements of the) social innovation can help to solve this issue. Research can support these solutions through development of tailored strategies and adequate business models for upscaling and replication, and development of

appropriate forms of cooperation with local governments or businesses. Research can further give insight in how to deal with the question of whether the complete social innovation could be upscaled or only some parts of it, and how and when this should be done.

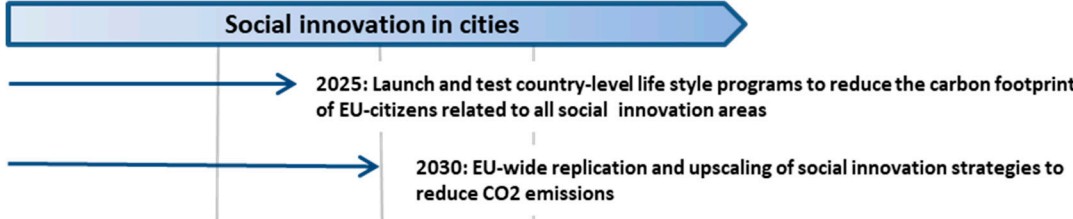

**Figure 3.** Key identified actions on social innovation for decarbonization in EU cities [3].

## 4. Conclusions: From Low Carbon Achievements to Zero-Carbon Cities in 2050

While the recommendations presented above are categorized into three pillars, they are highly interconnected and will all be needed to achieve zero-carbon cities. A system level approach, combining all areas of innovation listed above, will be needed to move from localized low carbon achievements to zero-carbon cities. This will involve many actors and diverse actions. First, strong city governance and vision will be needed. Clear targets and strategies will be needed to achieve the vision. For the transition to happen, citizens' buy-in and engagement will be crucial. All zero-carbon technology solutions will have to be tailored to the local context—and combined in "smart city" concepts. Electricity consumed in cities will need to be zero-carbon, therefore this challenge will also depend on the decarbonization happening in national power systems. Cities will also have to influence the power mix by locally producing renewable electricity. Transport and heating will need to become fully decarbonized as well—with a mix of renewable solutions and maximized internal flows. Waste will need to be minimized—and a circular economy realized. In summary, for zero-carbon cities, there will be no single "silver bullet" solution, but all solutions listed above will need to be used in conjunction and tailored to the local context [3].

Furthermore, a climate action in cities embeds a number of challenges that span across sectors. Climate policies can interact and have synergies (or trade-offs) with many development goals [49,50]. As examples, policies to improve livability and health outcomes in cities can also result in decarbonization, and vice versa. A clear example for this is the city of Barcelona case (Table 1), in which policies targeted at diminishing local air pollution also affected decarbonization outcomes. Furthermore, climate action in EU cities needs to be harmonized with other priorities, such as fighting energy poverty in cities [51]. A holistic approach to climate action in cities will, thus, be needed to capture the co-benefits of climate actions with other sustainability aspects. This includes planning climate mitigation and adaptation efforts in conjunction, and embedding nexus approaches that encompass several systems [49,52].

The R&I efforts listed above should also not treat cities in isolation. A large share of the connection between urban activities and both climate adaptation and mitigation run through city supply chains beyond city borders. "Embedded" emissions of imported goods are argued to be important to consider in city GHG inventories, along with subsequent mitigation efforts [53]. At the same time, these material/resource flows are increasingly vulnerable to the impacts of climate change, and need to be considered in climate adaptation planning [54]. Only scattered policies and research programs address the issue of "carbon leakage" of cities, even if it estimated that 12% to 35% of the European Union's consumption-based GHG emissions occur abroad [55]. That is particularly important in cities, as they are centers for the demand of products and materials.

This paper calls for selected R&I actions where higher education institutions can make a difference to the challenge of decarbonizing cities. The decarbonization challenge will require interdisciplinary and transdisciplinary science through broad cooperation between technical, economic, and social sciences, which poses a big challenge for scientists [56]. In interdisciplinary science there is integration

of knowledge and interaction between disciplinary scientists by which a better or new understanding of the issue is developed. Transdisciplinary science goes beyond interdisciplinarity, and includes other forms of knowledge derived from a wide range of stakeholders. Both forms of cross-disciplinary science require sound processes for knowledge sharing, interaction, and knowledge production and an adequate, highly knowledgeable mediation of the inter- or transdisciplinary process, as the interaction can sometimes become heated. Research should uncover the process requirements. "Living labs" engaging every actor, from citizens to academia, local businesses and the municipality, could be created in cities to test innovation in practice.

Universities, other higher education institutions (HEI), and educational and research systems in general will need to step up to the challenge. There is a need to train broadly educated or well-experienced researchers in the facilitation of the inter- and transdisciplinary processes. HEI can educate the new generation of professionals that are familiar with low-carbon transition challenges and solutions, and in particular, are able to envision and implement solutions across several sectors in the context of multi-actor governance. Universities can also build capacity in governments, and engage in outreach at schools or public events. In addition, universities can themselves lead by example and demonstrate low- and zero-carbon solutions by initiating high-visibility flagship projects that are also used for research.

Finally, multi-actor partnerships and networks, such as the Viable Cities project in Sweden [57], joining local authorities, HEI, companies, and others, can gather and expand the knowledge base for the zero-carbon transition in cities. Such partnerships can find and demonstrate innovative solutions, inform adaptive decision-making processes, or help collect data to monitor the implementation of zero-carbon projects in cities.

**Author Contributions:** F.F.N. coordinated inputs from the other authors, designed the study and wrote the paper. A.S. supported the design of the paper, led the writing of the social innovation (Section 3.3) and reviewed the paper. R.E.E. supported the design of the paper, wrote parts of the introductory and conclusion sections, and reviewed the paper. E.T. supported the design of the paper, led the writing of the governance (Section 3.2), and reviewed the paper.

**Funding:** The work presented in this paper was funded by the European Commission under contract number no. 642242 (https://deeds.eu/).

**Acknowledgments:** The authors acknowledge the High Level Panel for their collaboration on the Final Report of the High-Level Panel of the European Decarbonization Pathways Initiative, to which this study contributed. The research in this paper benefited from collaboration with the members of the DEEDS consortium. The authors also thank the collaborators in the city of Stockholm, and in particular former mayor of Stockholm Karin Wanngård, Björn Hugosson, and Malin Parmander.

**Conflicts of Interest:** The authors declare no conflict of interest.

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
