# Peer review of "A Research and Innovation Agenda for Zero-Emission European Cities"

_sustainability, doi:10.3390/su11061692_

Round 1

Reviewer 1 Report

The paper has a lot of good background and references but it lacks structure and argument. The intro doesn't make it abundantly clear what the researchers are trying to do - what the research question/aim is. Intro needs to provide a clear aim and finish with an overview of the structure of the paper so the readers know where you are taking them.

A lot seems to ride on Table 1 to show that cities are different. It's obvious that each city is different. I'm not sure why these three cities are chosen specifically? If the whole paper was a deeper analysis about these three cities, it might make more sense, but this seems to be the only time it's used. The Methods section is very weak. The results section is not really 'results' as you haven't got a method. It's more of a discussion. This 'results' section also needs more sub-headings or structure. It seems to bounce around a lot and I'm not sure what the argument being made there is. You have a lot of great general content and references, but the flow and argument needs work.

Your discussion (which I think is your conclusion?) is full of recommendations that aren't that novel. You need a proper conclusion.

A few very minor grammar errors.

Author Response

Thank you for your valuable comments that started a comprehensive review of the paper. The answers to each of the comments can be found in the attached file.

Reviewer 2 Report

This paper reviews some literature and summarized three key pillars in research and innovation agenda for achieving zero-emission for EU cities. Although we need to know what research items to focus on for achieving zero-emission for cities, this paper has not been clear on the scope of literature and the contribution to existing fields.

Question on method: what types of literature have been reviewed? Please explain this clearly, along with criteria of choosing literature. How many studies have been reviewed? This is basic information for readers to evaluate the scope of a review piece.

The concepts are overall the place. Smart city, circular economy, and innovation in technology are categorized as three key items in innovative technology and integration. But they are closely related and it is confusing about what the essence is here.

Why social innovation is separated from governance innovation? Do the authors thinking governance does not need citizens’ participation? The contribution of this paper to understand what we need for achieving zero-emission cities is not clear. What the authors presented is to combine several selected items together and claim it as a research and innovation agenda. Although the authors have cited many studies, the difference between many concepts is merely discussed and not clearly defined. Having this level of confusion, it is not useful. 

Author Response

(The authors gave the same response as above.)

Round 2

Reviewer 2 Report

The authors have provided more detail on methodology in this version, which helps to define the scope of this review. This is very helpful.

Author Response

Thank you for your valuable comments. We agree that an extended methods section was necessary for publication.